# Root Mean Square Layer Normalization

**Biao Zhang**[1]     **Rico Sennrich**[2,1]
[1]School of Informatics, University of Edinburgh
[2]Institute of Computational Linguistics, University of Zurich
B.Zhang@ed.ac.uk, sennrich@cl.uzh.ch

## Abstract

Layer normalization (LayerNorm) has been successfully applied to various deep neural networks to help stabilize training and boost model convergence because of its capability in handling re-centering and re-scaling of both inputs and weight matrix. However, the computational overhead introduced by LayerNorm makes these improvements expensive and significantly slows the underlying network, e.g. RNN in particular. In this paper, we hypothesize that re-centering invariance in LayerNorm is dispensable and propose root mean square layer normalization, or *RMSNorm*. RMSNorm regularizes the summed inputs to a neuron in one layer according to root mean square (RMS), giving the model re-scaling invariance property and implicit learning rate adaptation ability. RMSNorm is computationally simpler and thus more efficient than LayerNorm. We also present partial RMSNorm, or *pRMSNorm* where the RMS is estimated from $p\%$ of the summed inputs without breaking the above properties. Extensive experiments on several tasks using diverse network architectures show that RMSNorm achieves comparable performance against LayerNorm but reduces the running time by 7%~64% on different models. Source code is available at `https://github.com/bzhangGo/rmsnorm`.

## 1 Introduction

How to train deep neural networks efficiently is a long-standing challenge. To accelerate model convergence, Ba et al. [3] propose the layer normalization (LayerNorm) which stabilizes the training of deep neural networks by regularizing neuron dynamics within one layer via mean and variance statistics. Due to its simplicity and requiring no dependencies among training cases, LayerNorm has been widely applied to different neural architectures, which enables remarkable success on various tasks ranging from computer vision [19, 26], speech recognition [37] to natural language processing [31, 35]. In some cases, LayerNorm was found to be essential for successfully training a model [6]. Besides, the decoupling from batch-based samples endows LayerNorm with the superiority over batch normalization (BatchNorm) [12] in handling variable-length sequences using RNNs.

Unfortunately, the incorporation of LayerNorm raises computational overhead. Although this is negligible to small and shallow neural models with few normalization layers, this problem becomes severe when underlying networks grow larger and deeper. As a result, the efficiency gain from faster and more stable training (in terms of number of training steps) is counter-balanced by an increased computational cost per training step, which diminishes the net efficiency, as show in Figure 1. One major feature of LayerNorm that is widely regarded as contributions to the stabilization is its re-centering invariance property: the summed inputs after LayerNorm remain intact when the inputs or weight matrix is shifted by some amount of noise. We argue that this mean normalization does not reduce the variance of hidden states or model gradients, and hypothesize that it has little impact on the success of LayerNorm.

In this paper, we propose root mean square layer normalization (RMSNorm), which regularizes the summed inputs to a neuron in one layer with the root mean square (RMS) statistic alone.

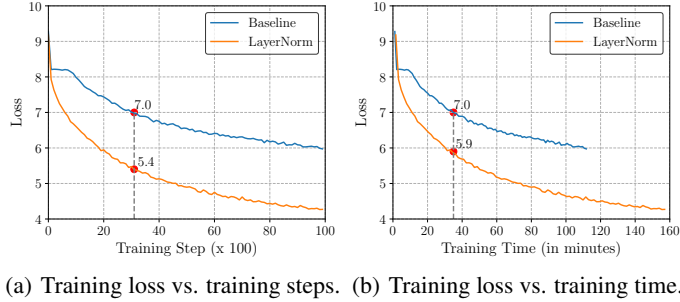

(a) Training loss vs. training steps.     (b) Training loss vs. training time.

Figure 1: Training procedure of a GRU-based RNNSearch [4] for the first 10k training steps. *Baseline* means the original model without any normalization. When the Baseline training loss arrives at 7.0, the loss of LayerNorm reaches 5.4 after the same number of training steps 1(a), but only 5.9 after the same training time 1(b).

RMSNorm reduces the amount of computation and increases efficiency over LayerNorm. Despite the simpler formulation, the RMS normalizer helps stabilize the magnitude of layer activations, ensuring invariance to the re-scaling of both weights and datasets. We also show the possibility of estimating RMS on a subset of the summed inputs, maintaining this invariance property. Assuming that the summed inputs have an independent identically distributed structure, we propose partial RMSNorm, where only the first $p\%$ summed inputs are utilized for RMS estimation.

We thoroughly examine our model on various tasks, including machine translation, image classification, image-caption retrieval and question answering. Experimental results show that across different models, RMSNorm yields comparable performance against LayerNorm but shows superiority in terms of running speed with a speed-up of 7%~64%. When estimating the RMS with partial (6.25%) summed inputs, $p$RMSNorm achieves competitive performance compared to RMSNorm.

## 2 Related Work

One bottleneck deep neural networks have been hypothesized to suffer from is the *internal covariate shift* issue [27], where a layer's input distribution changes as previous layers are updated, which significantly slows the training.[1] One promising direction to solve this problem is normalization. Ioffe and Szegedy [12] introduce batch normalization (BatchNorm) to stabilize activations based on mean and variance statistics estimated from each training mini-batch. Unfortunately, the reliance across training cases deprives BatchNorm of the capability in handling variable-length sequences, though several researchers develop different strategies to enable it in RNNs [16, 8]. Instead, Salimans and Kingma [22] propose weight normalization (WeightNorm) to reparameterize weight matrix so as to decouple the length of weight vectors from their directions. Ba et al. [3] propose layer normalization which differs from BatchNorm in that statistics are directly estimated from the same layer without accessing other training cases. Due to its simplicity and effectiveness, LayerNorm has been successfully applied to various deep neural models, and achieves state-of-the-art performance on different tasks [19, 37, 31, 6].

These studies pioneer the research direction that integrates normalization as a part of the model architecture. This paradigm ensures encouraging performance by shortening model convergence but at the cost of consuming more time for each running step. To improve efficiency, Arpit et al. [2] employ a data-independent method to approximately estimate mean and variance statistics, thus avoiding calculating batch statistics. Ioffe [11] propose batch renormalization so as to reduce the dependence of mini-batches in BatchNorm. Ulyanov et al. [30] replace batch normalization with instance normalization for image generation. Hoffer et al. [10] and Wu et al. [33] observe that $l1$-norm can act as an alternative of variance in BatchNorm with the benefit of fewer nonlinear operations and higher computational efficiency. Nevertheless, all these work still follow the original normalization structure and utilize mean statistic estimated from the whole summed inputs to handle re-centering invariance.

Different from these related work, the proposed RMSNorm modifies the normalization structure by removing the re-centering operation and regularizing the summed inputs with RMS alone. Our model only maintains the re-scaling invariance property which we find can be inherited when the RMS is estimated from only subset of the summed inputs, partially inspired by the group normalization [34]. As a side effect, our model reduces the computational overhead and increases efficiency. Recently, Zhang et al. [36] show that with careful initialization, residual networks can be trained as stable as those with normalization. However, the approach mainly aims at improving residual networks and can not be freely switched without modifying all initialization layers. Besides, it is not trivial to be adapted to other general neural networks, such as RNNs where model depth expands along the variable sequence length. By contrast, our model is simple, effective and can be used as a drop-in replacement of LayerNorm.

## 3   Background

We briefly review LayerNorm in this section based on a standard feed-forward neural network. Given an input vector $\mathbf{x} \in \mathbb{R}^m$, a feed-forward network projects it into an output vector $\mathbf{y} \in \mathbb{R}^n$ through a linear transformation followed by a non-linear activation as follows:

$$a_i = \sum_{j=1}^{m} w_{ij} x_j, \quad y_i = f\left(a_i + b_i\right), \tag{1}$$

where $\mathbf{w}_i$ is weight vector to the $i$-th output neuron, $b_i$ is bias scalar which is usually initialized by 0, and $f(\cdot)$ is an element-wise non-linear function. $\mathbf{a} \in \mathbb{R}^n$ denotes the weight-summed inputs to neurons, which is also the target of normalization.

This vanilla network might suffer from *internal covariate shift* issue [12], where a layer's input distribution changes as previous layers are updated. This could negatively affect the stability of parameters' gradients, delaying model convergence. To reduce this shift, LayerNorm normalizes the summed inputs so as to fix their mean and variance as follows:

$$\bar{a}_i = \frac{a_i - \mu}{\sigma} g_i, \quad y_i = f\left(\bar{a}_i + b_i\right), \tag{2}$$

where $\bar{a}_i$ is the $i$-th value of vector $\bar{\mathbf{a}} \in \mathbb{R}^n$, which acts as the normalized alternative of $a_i$ for layer activation. $\mathbf{g} \in \mathbb{R}^n$ is the gain parameter used to re-scale the standardized summed inputs, and is set to 1 at the beginning. $\mu$ and $\sigma^2$ are the mean and variance statistic respectively estimated from raw summed inputs $\mathbf{a}$:

$$\mu = \frac{1}{n} \sum_{i=1}^{n} a_i, \quad \sigma = \sqrt{\frac{1}{n} \sum_{i=1}^{n} (a_i - \mu)^2}. \tag{3}$$

Thus, LayerNorm forces the norm of neurons to be decoupled from the inputs and weight matrix.

## 4   RMSNorm

A well-known explanation of the success of LayerNorm is its re-centering and re-scaling invariance property. The former enables the model to be insensitive to shift noises on both inputs and weights, and the latter keeps the output representations intact when both inputs and weights are randomly scaled. In this paper, we hypothesize that the re-scaling invariance is the reason for success of LayerNorm, rather than re-centering invariance.

We propose RMSNorm which only focuses on re-scaling invariance and regularizes the summed inputs simply according to the root mean square (RMS) statistic:

$$\bar{a}_i = \frac{a_i}{\text{RMS}(\mathbf{a})} g_i, \quad \text{where } \text{RMS}(\mathbf{a}) = \sqrt{\frac{1}{n} \sum_{i=1}^{n} a_i^2}. \tag{4}$$

Intuitively, RMSNorm simplifies LayerNorm by totally removing the mean statistic in Eq. (3) at the cost of sacrificing the invariance that mean normalization affords. When the mean of summed inputs is zero, RMSNorm is exactly equal to LayerNorm. Although RMSNorm does not re-center

| | Weight matrix re-scaling | Weight matrix re-centering | Weight vector re-scaling | Dataset re-scaling | Dataset re-centering | Single training case re-scaling |
|---|---|---|---|---|---|---|
| BatchNorm | ✓ | ✗ | ✓ | ✓ | ✓ | ✗ |
| WeightNorm | ✓ | ✗ | ✓ | ✗ | ✗ | ✗ |
| LayerNorm | ✓ | ✓ | ✗ | ✓ | ✗ | ✓ |
| RMSNorm | ✓ | ✗ | ✗ | ✓ | ✗ | ✓ |
| $p$RMSNorm | ✓ | ✗ | ✗ | ✓ | ✗ | ✓ |

Table 1: Invariance properties of different normalization methods. "✓" indicates invariant, while "✗" denotes the opposite.

the summed inputs as in LayerNorm, we demonstrate through experiments that this property is not fundamental to the success of LayerNorm, and that RMSNorm is similarly or more effective.

RMS measures the quadratic mean of inputs, which in RMSNorm forces the summed inputs into a $\sqrt{n}$-scaled unit sphere. By doing so, the output distribution remains regardless of the scaling of input and weight distributions, benefiting the stability of layer activations. Although Euclidean norm which only differs from RMS by a factor of $\sqrt{n}$ has been successfully explored [22], we empirically find that it does not work for layer normalization. We hypothesize that scaling the sphere with the size of the input vector is important because it makes the normalization more robust across vectors of different size. As far as we know, the idea of employing RMS for neural network normalization has not been investigated before.

## 4.1 Invariance Analysis

Invariance measures whether model output after normalization changes highly in accordance with its input and weight matrix. Ba et al. [3] show that different normalization methods reveal different invariance properties, which contributes considerably to the model's robustness. In this section, we theoretically examine the invariance properties of RMSNorm.

We consider the following general form of RMSNorm:

$$\mathbf{y} = f\left(\frac{\mathbf{Wx}}{\text{RMS}(\mathbf{a})} \odot \mathbf{g} + \mathbf{b}\right),\tag{5}$$

where $\odot$ denotes element-wise multiplication. Our main results are summarized in Table 1. RMS-Norm is invariant to both weight matrix and input re-scaling, because of the following linearity property of RMS:

$$\text{RMS}(\alpha\mathbf{x}) = \alpha\text{RMS}(\mathbf{x}),\tag{6}$$

where $\alpha$ is a scale value. Suppose the weight matrix is scaled by a factor of $\delta$, i.e. $\mathbf{W}' = \delta\mathbf{W}$, then this change does not affect the final layer output:

$$\mathbf{y}' = f\left(\frac{\mathbf{W}'\mathbf{x}}{\text{RMS}(\mathbf{a}')} \odot \mathbf{g} + \mathbf{b}\right) = f\left(\frac{\delta\mathbf{Wx}}{\delta\text{RMS}(\mathbf{a})} \odot \mathbf{g} + \mathbf{b}\right) = \mathbf{y}.\tag{7}$$

By contrast, if the scaling is only performed on individual weight vectors, this property does not hold anymore as different scaling factors break the linearity property of RMS. Similarly, if we enforce a scale on the input with a factor of $\delta$, i.e. $\mathbf{x}' = \delta\mathbf{x}$, the output of RMSNorm remains through an analysis analogous to that in Eq. 7. We can easily extend the equality to batch-based inputs as well as the whole dataset. Therefore, RMSNorm is invariant to the scaling of its inputs.

The main difference to LayerNorm is that RMSNorm is not re-centered and thus does not show similar linearity property for variable shifting. It is not invariant to all re-centering operations.

## 4.2 Gradient Analysis

The above analysis only considers the effect of scaling inputs and the weight matrix on the layer output. In a general setting, however, a RMSNorm-enhanced neural network is trained via standard stochastic gradient descent approach, where the robustness of model gradient is very crucial to parameters' update and model convergence (see also Santurkar et al. [23] who argue that the success of normalization methods does not come from the added stability to layer inputs, but due to increased smoothness of the optimization landscape). In this section, we investigate the properties of model gradients in RMSNorm.

Given a loss function $\mathcal{L}$, we perform back-propagation through Eq. (4) to obtain the gradient with respect to parameters $\mathbf{g}, \mathbf{b}$ as follows:

$$\frac{\partial \mathcal{L}}{\partial \mathbf{b}} = \frac{\partial \mathcal{L}}{\partial \mathbf{v}}, \quad \frac{\partial \mathcal{L}}{\partial \mathbf{g}} = \frac{\partial \mathcal{L}}{\partial \mathbf{v}} \odot \frac{\mathbf{Wx}}{\text{RMS}(\mathbf{a})}, \tag{8}$$

where $\mathbf{v}$ is short for the whole expression inside $f(\cdot)$ in Eq. (4), and $\partial \mathcal{L}/\partial \mathbf{v}$ is the gradient back-propagated from $\mathcal{L}$ to $\mathbf{v}$. Both gradients $\partial \mathcal{L}/\partial \mathbf{b}$ and $\partial \mathcal{L}/\partial \mathbf{g}$ are invariant to the scaling of inputs $\mathbf{x}$ and the weight matrix $\mathbf{W}$ (in the case of $\partial \mathcal{L}/\partial \mathbf{g}$ because of the linearity property in Eq. (6)). Besides, the gradient of $\mathbf{g}$ is proportional to the normalized summed inputs, rather than raw inputs. This powers the stability of the magnitude of $\mathbf{g}$.

Unlike these vector parameters, the gradient of the weight matrix $\mathbf{W}$ is more complicated due to the quadratic computation in RMS. Formally,

$$\frac{\partial \mathcal{L}}{\partial \mathbf{W}} = \sum_{i=1}^{n} \left[ \mathbf{x}^T \otimes \left( \text{diag} \left( \mathbf{g} \odot \frac{\partial \mathcal{L}}{\partial \mathbf{v}} \right) \times \mathbf{R} \right) \right]_i, \text{where } \mathbf{R} = \frac{1}{\text{RMS}(\mathbf{a})} \left( \mathbf{I} - \frac{(\mathbf{Wx})(\mathbf{Wx})^T}{n\text{RMS}(\mathbf{a})^2} \right), \tag{9}$$

$\text{diag}(\cdot)$ denotes the diagonal matrix of input, $\otimes$ denotes the Kronecker product, and "$\mathbf{I}$" indicates identity matrix. For clarity, we explicitly use "$\times$" to represent matrix multiplication. The matrix term $\mathbf{R}$ associates the gradient of $\mathbf{W}$ with both inputs $\mathbf{x}$ and weight matrix $\mathbf{W}$. With a thorough analysis, we can demonstrate that this term is negatively correlated with both input and weight matrix scaling. After assigning a scale of $\delta$ to either input $\mathbf{x}$ ($\mathbf{x}' = \delta \mathbf{x}$) or weight matrix ($\mathbf{W}' = \delta \mathbf{W}$), we have

$$\mathbf{R}' = \frac{1}{\delta\text{RMS}(\mathbf{a})} \left( \mathbf{I} - \frac{(\delta\mathbf{Wx})(\delta\mathbf{Wx})^T}{n\delta^2\text{RMS}(\mathbf{a})^2} \right) = \frac{1}{\delta}\mathbf{R}. \tag{10}$$

If we put the scaled term $\mathbf{R}'$ back into Eq. (9), we can easily prove that the gradient $\partial \mathcal{L}/\partial \mathbf{w}$ is invariant to input scaling, but keeps the negative correlation with weight matrix scaling. Reducing the sensitivity of gradient $\partial \mathcal{L}/\partial \mathbf{w}$ to the scaling of inputs ensures its smoothness and improves the stability of learning. On the other hand, the negative correlation acts as an implicit learning rate adaptor and dynamically controls the norm of gradients which avoids large-norm weight matrix and improves model convergence.

## 5   $p$**RMSNorm**

The re-scaling invariance property of RMSNorm ascribes to the linearity property of RMS. Considering that neurons in one layer often have independent identically distributed structure, we argue that the RMS can be estimated on a subset of these neurons rather than all of them. We propose partial RMSNorm ($p$RMSNorm). Given the unnormalized input $\mathbf{a}$, $p$RMSNorm infers the RMS statistic from first-$p\%$ elements of $\mathbf{a}$: $\overline{\text{RMS}}(\mathbf{a}) = \sqrt{\frac{1}{k}\sum_{i=1}^{k} a_i^2}$, where $k = \lceil n \cdot p \rceil$ denotes the number of elements used for RMS estimation. The linearity property still holds for $\overline{\text{RMS}}$ as in Eq. (6), which indicates $p$RMSNorm shares the same invariance properties as RMSNorm as shown in Table 1.

$\overline{\text{RMS}}$ is a biased estimation of the RMS which is often inaccurate. Though theoretically $p$RMSNorm approximates to RMSNorm, we observe gradient instability where the gradient tends to explode with small $m$. In practice, however, models with $p$RMSNorm can succeed in satisfactory convergence with a partial ratio of 6.25%.

## 6   Experiments

To test the efficiency of layer normalization across different implementations, we perform experiments with Tensorflow [1], PyTorch [20] and Theano [29]. We add RMSNorm to different models, comparing against an unnormalized baseline and LayerNorm. These models are based on diverse architectures, covering different RNN variants, convolutional and self-attentional models, and various activations (such as sigmoid, tanh, and softmax), with initialization ranging from uniform, normal, orthogonal with different initialization ranges or variances. Unless otherwise noted, all speed-related statistics are measured on one TITAN X (Pascal). Reported time is averaged over 3 runs. We also list the standard deviation of these three runs.

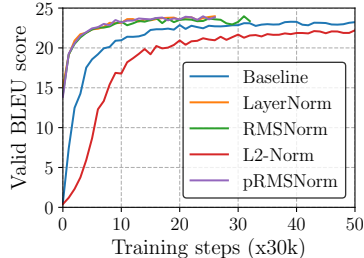

Figure 2: SacreBLEU score on newstest2013 for the RNNSearch. Models are implemented according to *Nematus* [25] in Tensorflow.

| Model | Test14 | Test17 | Time |
|---|---|---|---|
| Baseline | 21.7 | 23.4 | 399±3.40s |
| LayerNorm | 22.6 | 23.6 | 665±32.5s |
| L2-Norm | 20.7 | 22.0 | 482±19.7s |
| RMSNorm | 22.4 | **23.7** | 501±11.8s (24.7%) |
| *p*RMSNorm | **22.6** | 23.1 | 493±10.7s (25.9%) |

Table 2: SacreBLEU score on newstest2014 (Test14) and newstest2017 (Test17) for RNNSearch using Tensorflow-version Nematus. "*Time*": the time in second per 1k training steps. We set $p$ to 6.25%. We highlight the best results in bold, and show the speedup of RMSNorm against Layer-Norm in bracket.

## 6.1 Machine Translation

Machine translation aims at transforming a sentence from one (source) language to another (target) language. We focus on neural machine translation based on an attention-enhanced encoder-decoder framework. We train two different models, a GRU-based RNNSearch [4] and a self-attention based neural Transformer [31] on WMT14 English-German translation task. More details about the experimental settings as well as comparison with WeightNorm are listed in Appendix A.1

We first experiment with RNNSearch. Normalization is added to the recurrent connections and feedforward layers. Apart from RNNSearch without any normalization (Baseline) and with Layer-Norm, we also compare against the same model equipped with L2-Norm (i.e. replacing RMS with L2-Norm), which has been observed to improve lexical selection [18].

Figure 2 illustrates the evolution of BLEU score on our development set after every 30k training steps, and Table 2 summarizes the test results. In short, both LayerNorm and RMSNorm outperform the Baseline by accelerating model convergence: they reduce the number of training steps until convergence by about 50%, and improve test accuracy, with RMSNorm being comparable to LayerNorm. This supports our hypothesis that re-scaling invariance is the core property of LayerNorm, and that RMSNorm is an effective substitute. Our results with L2-Norm show that it fails to improve the model.[2] Results in Table 2 highlight the challenge that RNN with LayerNorm in Tensorflow suffers from serious computational inefficiency, where LayerNorm is slower than the Baseline by about 67%. In this respect, RMSNorm performs significantly better, improving upon LayerNorm by ~25%.

Table 3 further lists translation results of different models implemented in Theano and Pytorch. Overall, RMSNorm yields comparable translation quality compared with LayerNorm but incurs less computational overhead, outperforming LayerNorm with speedups ranging from 11%~34%. In addition, we observe that though in theory the amount of computation in *p*RMSNorm is less than that in RMSNorm, *p*RMSNorm ($p = 6.25\%$) sometimes tends to be slower. We ascribe this to the non-optimal implementation of tensor slicing operation in these computational frameworks, which can be improved with specific low-level coding.

In *p*RMSNorm, the partial ratio $p$ directly controls the accuracy of estimated RMS, thereby affecting the stability of model training. Figure 3 shows the effect of $p$ on model performance. Surprisingly, we find that the scale of $p$ has little influence on the final translation quality in RNNSearch: using a small ratio does not significantly degenerate BLEU score. We set $p$ to 6.25% for all following experiments.

We also experiment with Transformer, which is based on self-attention, avoiding recurrent connections and allowing a higher degree of parallelization. Still, layer normalization is an important part of the architecture. We use an in-house Tensorflow implementation of the Transformer, and employ the base setting as in [31] with all models trained for 300K steps. We treat Transformer with no normalization as our Baseline, and compare RMSNorm-enhanced Transformer with LayerNorm-equipped Transformer. Table 4 shows the results, from which we observe the importance of normalization for Transformer, without which training fails. RMSNorm achieves BLEU scores comparable to LayerNorm, and yields a speedup of 7%~9%. Compared with RNNSearch, the relative cost of

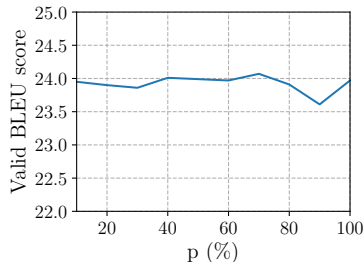

Figure 3: SacreBLEU score on new-stest2013 (devset) for the RNNSearch with $p$RMSNorm. We use Tensorflow-version Nematus, and change $p$ by a step size of 10%.

Table 3: SacreBLEU score on newstest2014 (Test14) and new-stest2017 (Test17) for RNNSearch. "*Th*": Theano-version Nematus, "*Py*": an in-house PyTorch-based RNNSearch.

|  | Model | Test14 | Test17 | Time |
|---|---|---|---|---|
| *Th* | Baseline | 21.8 | 22.9 | 596±20.8s |
|  | LayerNorm | 22.3 | 23.8 | 988±1.10s |
|  | RMSNorm | 22.5 | 23.2 | 652±24.1s (34.0%) |
|  | $p$RMSNorm | **22.7** | **24.0** | 658±17.9s (33.4%) |
| *Py* | Baseline | 22.7 | **24.7** | 427±6.50s |
|  | LayerNorm | 23.2 | 24.3 | 857±17.2s |
|  | RMSNorm | 22.9 | 24.5 | 763±16.2s (11.0%) |
|  | $p$RMSNorm | **23.2** | 24.6 | 754±36.1s (12.0%) |

| Model | Test14 | Test17 | Time |
|---|---|---|---|
| Baseline | - | - | 210±0.23s |
| LayerNorm | 26.6 | 27.7 | 248±1.31s |
| RMSNorm | **26.8** | 27.7 | 231±0.04s (6.9%) |
| $p$RMSNorm | 26.5 | **27.8** | 225±1.63s (9.3%) |

Table 4: SacreBLEU score on newstest2014 (Test14) and newstest2017 (Test17) for the Transformer. "*Time*": the time in second per 1k training steps, which is measured using Tesla V100. "-" indicates that we fail to train this model and BLEU score is 0.

| Model |  | 1 | 2 | 3 | 4 | ALL |
|---|---|---|---|---|---|---|
| Baseline | M | -2.60 | -1.19 | -1.43 | -1.53 | -1.60 |
|  | S | 7.35 | 2.33 | 2.61 | 2.73 | 3.04 |
| LayerNorm | M | -0.43 | -0.48 | -0.50 | -0.50 | -0.51 |
|  | S | 1.19 | 1.51 | 1.51 | 1.51 | 1.51 |
| RMSNorm | M | -0.40 | -0.60 | -0.69 | -0.74 | -0.73 |
|  | S | 1.27 | 1.51 | 1.50 | 1.49 | 1.50 |

Table 5: Mean (*M*) and standard deviation (*S*) statistics estimated on the hidden-to-hidden mapping of decoder-part GRU cell in RNNSearch model. We use the newstest2013 dataset. *ALL*: the statistics averaged across all token positions. Numbers *1,2,3,4* indicate the statistic estimated for specific token positions.

normalization is lower because there are significantly fewer sequential normalization operations in Transformer.

**Effect of Normalization on Mean and Standard Deviation** Table 5 shows the distribution of mean and standard deviation of hidden representations across token positions for an RNNSearch model. Mean and standard deviation are unstable in the baseline, as observed by Ba et al. [3]. Due to their normalization properties, both RMSNorm and LayerNorm stabilize standard deviation. Although the mean in RMSNorm is not normalized, in practice it is more stable than the mean of the baseline. This supports our hypothesis that RMSNorm stabilizes recurrent activations without the need to explicitly normalize the mean.

**On the Robustness of RMSNorm** One remaining question is whether the re-centering operation in LayerNorm (which RMSNorm abandons) makes models more robust towards arbitrary weight/bias initializations. We perform an experiment on RNNSearch with Nematus in Tensorflow, and change the center of weight initialization to 0.2. Results in Figure 4 show that LayerNorm becomes very unstable with abnormal initialization, but RMSNorm is more robust (both underperform the original initialization). Our empirical evidence so far suggests that RMSNorm is similarly robust as LayerNorm, or more.

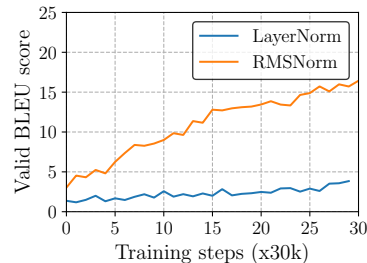

Figure 4: SacreBLEU score curve of Layer-Norm and RMSNorm on newstest2013 (devset) when the initialization center is 0.2.

## 6.2 CNN/Daily Mail Reading Comprehension

This reading comprehension task is a cloze-style question answering task, where models are required to answer a question regarding to a passage, and the answer is an anonymized entity from the passage [9]. We train a bidirectional attentive reader model proposed by Hermann et al. [9] on the CNN corpus. More details about the experimental settings are given in Appendix A.2. We compare RMSNorm with both LayerNorm and BatchNorm.

Figure 5 and Table 6 show the results. After normalizing RNN by BatchNorm with separate statistics for each time step in a sequence, both BatchNorm-LSTM and BatchNorm-Everywhere help speed up the convergence of training process. By contrast, LayerNorm and RMSNorm not only converge faster than BatchNorm, but also reach lower validation error rate, though $p$RMSNorm performs slightly

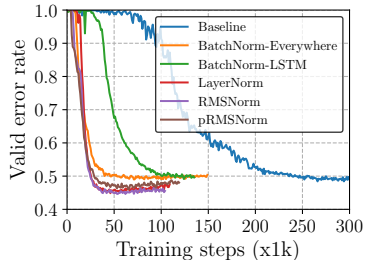

Figure 5: Error rate on validation set for the attentive reader model.

| Model | Time |
|---|---|
| Baseline | 315±6.30s |
| BatchNorm-Everywhere | 348±10.5s |
| BatchNorm-LSTM | 345±11.2s |
| LayerNorm | 392±5.70s |
| RMSNorm | 333±5.20s (15.1%) |
| $p$RMSNorm | 330±5.50s (15.8%) |

Table 6: Time in seconds per 0.1k training steps for the attentive reader model.

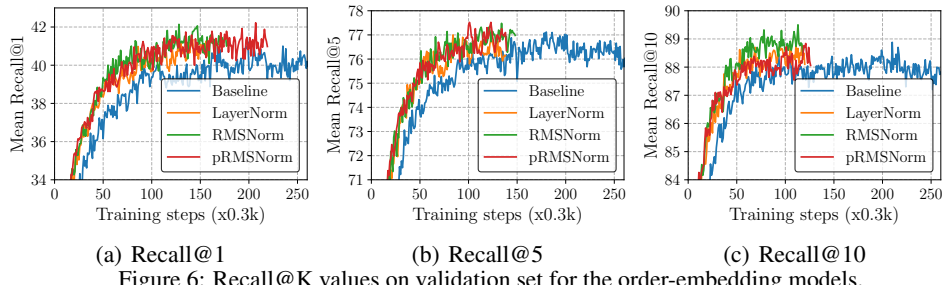

(a) Recall@1      (b) Recall@5      (c) Recall@10

Figure 6: Recall@K values on validation set for the order-embedding models.

worse than RMSNorm. Although in Figure 5 the performance of RMSNorm and LayerNorm is comparable, RMSNorm is around 15% faster than LayerNorm as shown in Table 6.[3]

## 6.3 Image-Caption Retrieval

Image-caption retrieval is a cross-modal task aiming at learning a joint embedding space of images and sentences, which consists of two sub-tasks: *image retrieval* and *caption retrieval*. The former ranks a set of images according to a query caption, and the latter ranks a set of captions based on a query image. We train an order-embedding model (OE) proposed by Vendrov et al. [32] on the Microsoft COCO dataset [17] using their public source code in Theano. Model details about experimental settings are provides in Appendix A.3. We compare RMSNorm with two models: one without any normalization (Baseline) and one with LayerNorm.

Figure 6 shows the R@K curve on validation set after every 300 training steps, and Table 7 lists the final test results. Across all these metrics, RMSNorm and LayerNorm consistently outperform the Baseline in terms of model convergence as shown in Figure 6. We observe that on the validation set, RMSNorm slightly exceeds LayerNorm with respect to recall value. For the final test results as

| Model | Time |
|---|---|
| Baseline | 2.11±0.047s |
| LayerNorm | 12.02±0.191s |
| RMSNorm | 7.12±0.207s (40.8%) |
| $p$RMSNorm | 4.34±0.168s (63.9%) |

Table 8: Time in seconds per 0.1k training steps for the order-embedding model.

shown in Table 7, both RMSNorm and LayerNorm improve the model performance, reaching higher recall values (except LayerNorm on R@5) and lower mean rank, though RMSNorm reveals better generalization than LayerNorm. Besides, results in Table 8 show that RMSNorm accelerates training speed by 40%∼64% compared with LayerNorm, highlighting better efficiency of $p$RMSNorm.

## 6.4 CIFAR-10 Classification

CIFAR-10 is a supervised image classification task, with 10 different classes. We train a modified version of the ConvPool-CNN-C architecture [15], and follow the same experimental protocol as Salimans and Kingma [22]. BatchNorm, LayerNorm, and WeightNorm are included for comparison. Training details are given in Appendix A.4.

Figure 9 and Table 10 show the results. Models enhanced with a normalization technique converge faster than Baseline, among which BatchNorm performs the best. Similar to previous observation [3],

| | Model | Caption Retrieval | | | | Image Retrieval | | | |
|---|---|---|---|---|---|---|---|---|---|
| | | **R@1** | **R@5** | **R@10** | **Mean r** | **R@1** | **R@5** | **R@10** | **Mean r** |
| Existing Work | Sym [32] | 45.4 | | 88.7 | 5.8 | 36.3 | | 85.8 | 9.0 |
| | OE + Baseline [32][†] | 46.7 | | 88.9 | 5.7 | 37.9 | | 85.9 | 8.1 |
| | OE + Baseline [3][‡] | 46.6 | 79.3 | 89.1 | 5.2 | 37.8 | 73.6 | 85.7 | 7.9 |
| | OE + LayerNorm [3] | 48.5 | **80.6** | 89.8 | **5.1** | 38.9 | 74.3 | 86.3 | 7.6 |
| This Work | OE + Baseline | 45.8 | 79.7 | 88.8 | 5.4 | 37.6 | 73.6 | 85.8 | 7.7 |
| | OE + LayerNorm | 47.9 | 79.5 | 89.2 | 5.3 | 38.4 | 74.6 | **86.7** | 7.5 |
| | OE + RMSNorm | **48.7** | 79.7 | 89.5 | 5.3 | **39.0** | **74.8** | 86.3 | 7.5 |
| | OE + $p$RMSNorm | 46.8 | 79.8 | **90.3** | 5.2 | 39.0 | 74.5 | 86.3 | **7.4** |

Table 7: Average R@K values across 5 test sets from Microsoft COCO. **R@K**: Recall @ K, higher is better. **Mean r**: mean rank, lower is better. The number in bold highlights the best result. [‡] denotes the reproduced results of [†].

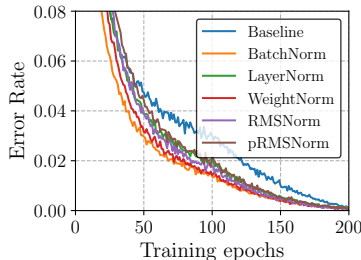

| Model | Test Error | Time |
|---|---|---|
| Baseline | 8.96% | 21±0.0s |
| BatchNorm | 8.25% | 38±0.0s |
| WeightNorm | 8.28% | 23±0.0s |
| LayerNorm | 10.49% | 39±0.4s |
| RMSNorm | 8.83% | 31±0.5s (20.5%) |
| $p$RMSNorm | 10.37% | 30±0.4s (23.1%) |

Table 10: Test error rate and time in seconds per training epoch for the ConvPool-CNN-C model. Time is measured with GeForce RTX 2080 Ti.

Table 9: Training error rate for the ConvPool-CNN-C model.

we also find that layer normalization works worse than BatchNorm and WeightNorm for image processing. Though LayerNorm outperforms Baseline by shorting model convergence, it fails to generalize to the test set, degenerating the test error by 1.53%. In contrast, RMSNorm shows better generalization, surpassing the Baseline by 0.013% and saving about 20.5% training time compared to LayerNorm. $p$RMSNorm gains further speedup of 2.6%, albeit at the cost of sacrificing test accuracy of 1.54%.

# 7 Conclusion and Future Work

This paper presents RMSNorm, a novel normalization approach that normalizes the summed inputs according to the RMS. RMSNorm preserves the re-scaling invariance property of LayerNorm but eschews the re-centering invariance property which contributes less to the model training. Compared with LayerNorm, models with RMSNorm suffers from less computational overhead. RMSNorm can be easily applied to different model architectures as a drop-in replacement of LayerNorm. Experiments on several NLP tasks show that RMSNorm is comparable to LayerNorm in quality, but accelerates the running speed. Actual speed improvements depend on the framework, hardware, neural network architecture and relative computational cost of other components, and we empirically observed speedups of 7%~64% across different models and implementations. Our efficiency improvement come from simplifying the computation, and we thus expect them to be orthogonal to other means of increasing training speed, such as low-precision arithmetic and GPU kernel fusion. We also experimented with $p$RMSNorm which estimates the RMS on a subset of the summed inputs. While theoretically faster, we did not consistently observe empirical speed improvements for $p$RMSNorm. We leave it to future work to investigate if the performance can be improved via code optimization.

In the future, we would like to take more analysis about the success behind RMSNorm. Inspired by recent success of $l1$-norm for BatchNorm, we will explore different norms for RMSNorm, and simplify other normalization techniques such as BatchNorm.

# Acknowledgments

We thank the reviewers for their insightful comments, and Antonio Valerio Miceli Barone for his support with weight normalization for MT. This project has received funding from the grant H2020-ICT-2018-2-825460 (ELITR) by the European Union. Biao Zhang also acknowledges the support of the Baidu Scholarship. This work has been performed using resources provided by the Cambridge Tier-2 system operated by the University of Cambridge Research Computing Service (http://www.hpc.cam.ac.uk) funded by EPSRC Tier-2 capital grant EP/P020259/1.

## Footnotes

[1]Note that the internal covariate shift is given as motivation by [12, 3]. Recent studies have proposed alternative explanations for the success of normalization, such as the uncontrollable growth of layer activations in unnormalized deep networks [5].

[2]We note that Nguyen and Chiang [18] only applied L2-Norm to the last layer, and treat the scaling factor as a hyperparameter. While not a replication of their experiment, we still found it worth testing L2-Norm as an alternative to LayerNorm.

[3]Notice that the implementation of BatchNorm is cuDNN-based, so time cost of BatchNorm in Table 6 can not be directly compared with others.

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
