[Supplementary Material]

# A  Appendix

## A.1  Machine Translation

We experiment on the WMT14 English-German translation task, where the training corpus consists of 4.5M aligned sentence pairs. We use newstest2013 as the development set for model selection, newstest2014 and newstest2017 as the test set. We evaluate translation quality with case-sensitive detokenized BLEU score reported by *sacrebleu* [21][4]. Byte pair encoding algorithm is applied to reduce out-of-vocabulary tokens with 32k merge operations [24]. All models are trained based on maximum log-likelihood relaxed by label smoothing with a factor of 0.1.[5]

For RNNSearch, we select training samples with maximum source/target length of 80. We set the embedding size and hidden size to be 512 and 1024, respectively. We apply Adam [13] to tune model parameters, with learning rate of $1e^{-4}$ and batch size of 80. We initialize weight parameters with standard normal distribution scaled by 0.01, except for square weight parameters which are handled with random orthogonal matrix.

For Transformer, we adopt the base setting as in [31]. We set the model size and FFN hidden size to be 512 and 2048, respectively, and use 8 attention heads. The learning rate is scheduled according to the inverse square root of running steps, with warmup steps of 4000. We organize each batch with 25000 source/target tokens, and train the model using Adam ($\beta_1 = 0.9, \beta_2 = 0.98$) [13]. We apply Xavier initialization to all weight parameters.

We also compare our model with WeightNorm. We perform experiments with RNNSearch using the WeightNorm implementation provided by Theano-version Nematus. Results in Figure 7 show that WeightNorm converges slower and requires more training steps. In addition, the overall translation quality of WeightNorm on testsets (21.7/23.5 on Test14/Test17, respectively) underperforms those of LayerNorm and ($p$)RMSNorm. We also attempted integrating WeightNorm into pytorch-based RNNSearch using the official API (*nn.utils.weight_norm*), but this led to out-of-memory problems.

Figure 7: SacreBLEU score curve over training steps on newstest2013 (devset) for the RNNSearch. Models are trained with *Nematus* in Theano.

## A.2  CNN/Daily Mail Reading Comprehension

The model is trained on the CNN corpus based on the public source code in Theano [8]. We adopt the *top4* setting, where each passage in the pre-processed dataset contains at most 4 sentences. For fair comparison with LayerNorm, we only employ RMSNorm within LSTM. We set hidden size of LSTM to be 240. Models are optimized via Adam optimizer [13] with a batch size of 64 and learning rate of $8e^{-5}$. Xavier initialization is adopted for all weight parameters with square weights further enforced orthogonality.

## A.3  Image-Caption Retrieval

In OE model, sentences are encoded through a GRU-based RNN [7] and images are represented by the output of a pretrained VGGNet [28]. OE treats the caption-image pairs as a two-level partial order, and trains the joint model using the pairwise ranking loss [14].

We adopt the *10crop* feature from VGGNet as image representation, and set word embedding size and GRU hidden size to be 300 and 1024 respectively. Xavier, Gaussian and Ortogonal initialization are used for different model parameters. All models are trained with Adam optimizer, with a batch size of 128 and learning rate of $1e^{-3}$. We employ Recall@K (R@K) values for evaluation, and report averaged results on five separate test sets (each consisting of 1000 images and 5000 captions) as our final test results.

### A.4 CIFAR-10 Classification

We apply layer normalization to the width and height dimensions of image representation, and perform gain scaling and bias shifting on the channel dimension. We train all models using Adam optimizer with a batch size of 100. Learning rate is set to 0.0003 for Baseline and 0.003 for others following [22]. We set $p$ to 12.5% for $p$RMSNorm, considering the small model size in this task.

## Footnotes

[4]Sacrebleu hash: BLEU+case.mixed+lang.en-de+numrefs.1+smooth.exp+test.wmt14+tok.13a+version.1.2.12 and BLEU+case.mixed+lang.en-de+numrefs.1+smooth.exp+test.wmt17+tok.13a+version.1.2.12.

[5]Note that there are minor differences between different frameworks, both in implementation details and setup, explaining performance differences between the baselines.