[Reviews · NeurIPS 2019]

Reviewer 1



The authors propose a new normalization scheme called RMSNorm. They provide an interesting (although somewhat straightforward) theoretical analysis, and empirically show benefits of the approach. Detailed comments follow: - The paper is quite nicely written, and it is in general easy to follow authors' line of thought. - Figure 1 doesn't seem to indicate that there is a large problem with LayerNorm, as the difference is quite small. Maybe choose a different example that better illustrates this issue. - In the related work, the authors mention "internal covariate shift" as a fact, while there is a lot of recent work that questions this explanation. Some the authors do mention (such as [21]), but some are not (e.g., Bjorck et al. "Understanding batch normalization", NeurIPS 2018). Would be good to add a discussion on ICS to reflect these recent works, and not consider the ICS as a fact. - In (2), should the second part use vector notation as done in (1)? Or should it be vice versa, (1) should use scalar notation instead? - Please never use "obviosuly" in any of your papers (or "clearly", or any other similar word). What is obvious to you may not be obvious to others. Rephrase this sentence, or you could even add a short equation explain it (although not really necessary). - In Fig 3, is the x-axis really in "%"? Doesn't seem to be, is the max value considered really just "1%"? - What is a difference between various versions of OE and OE+LayerNorm in Table 7? Several rows have the same caption. This should be explained and elaborated, it is quite confusing at the moment. - Typos: "cloze", "shorting" ** Comments after the author response ** Thank you for a very detailed response! I am staying with my aept recommendation.

Reviewer 2



This is mostly an engineering/empirical paper which simply explores an architecture modification. The paper is clearly written and the idea is fairly straight forward: the authors propose to use LayerNorm without mean centering. As such, the originality is limited. Still, there has recently been a proliferation of different network architectures and especially normalization techniques, so any possible simplification with only minimal performance losses should be welcomed by the community (in the spirit of Occam's razor). I'm not very familiar with the particular tasks the authors use to compare their method. There is a focus on language models, and this is likely because LayerNorm happens to provide good performance on these types of tasks. But since the paper brings only minimal theoretic contributions to the table, it would be helpful to compare performance on tasks which involve different types of architectures, as well as data (such as images, sound, video, etc.). EDIT: I thank the authors for their helpful clarifications. I’m still somewhat on the fence about this paper, it mainly representing a fix to an earlier method (it appears the authors of LayerNorm were missing an ablation experiment). However, overall the paper represents a net benefit to the community, which shouldn’t be ignored. I’ll update my score to 7.

Reviewer 3



ORIGINALITY: + The proposed normalization technique is original in the sense that the main difference in existing normalization techniques (batch, layer, group, instance..) differ only in the dimensions over which the activations are normalized. This paper proposes removing one of the typical steps in the normalization process in order to speed up training, which has been less well-studied - This work proposes dividing by the RMS statistic instead of standard deviation without hurting accuracy. Other works (for example, Santurkar et al.) experiment with scaling by different statistics, such as various l_p norms, without a loss in training accuracy. This work is not the first to suggest scaling the activations by a different statistic QUALITY: + The authors tested their technique on multiple deep learning frameworks (TensorFlow, PyTorch, Theano), which gives more support to their empirical results, as different implementations can have very different timing results + The authors tested their technique on multiple tasks and neural network architectures - The main hypothesis hypothesis is that the re-centering step in Layer Normalization is dispensable, and this is backed only by experimental results and could be a lot stronger with some theoretical justification - While the few experimental results show that there is no degradation of accuracy from not centering the activations, I am still not fully convinced that the centering step can be deemed unnecessary. For example, it is likely that the weights/biases of the networks in the paper are initialized such that the activations are roughly centered around zero already, and therefore the mean-centering step can be removed without seeing much of a difference in performance. An advantage of existing techniques such as Batch Normalization and Layer Normalization is that they are more robust to hyperparameter selection such as learning rate, but more importantly in this case, weight/bias initialization. It’s possible that the proposed technique may not work as well as existing techniques for arbitrary weight/bias initializations - Would be good to also compare to Weight Normalization, which is motivated by reducing computational overhead compared to Batch Normalization CLARITY: + The paper is generally clearly written and each section flows logically after the next. - Some minor details about the experiments are missing - for example, the learning rate/optimizer used for training is given for Daily Mail Reading and Image-Caption Retrieval but not Machine Translation. Weight initialization methods are missing entirely and would be useful to know for anyone trying to reimplement these results. Mention of the type of nonlinearity used in each network is also absent. SIGNIFICANCE: + Drop-in replacement for networks with Layer Normalization that can speed up training without reducing accuracy. Accuracy with RMSNorm/pRMSNorm is slightly better than Layer Normalization but these differences are small and it is unclear if these differences are within noise + Easy to implement in existing models, so easy for practitioners to use - Depending on the circumstances, practitioners often care about final accuracy and inference speed more than training speed. The proposed technique does not convincingly improve asymptotic accuracy and would improve inference speed over Layer Normalization, but not Batch Normalization if the inference-time parameters are ‘folded in’ to the preceding layer’s weights

[Author Response · NeurIPS 2019]

Dear Reviewers:

Thanks for all your insightful comments and constructive suggestions. We will correct all typos in our final version. We
first respond to a common concern:

*About generality of RMSNorm for different downstream tasks, model architec-*
*tures, and initializations:* We mainly experiment on language-related tasks,
because this is where the use of LayerNorm is most widespread. However, note
that our experiments show the effectiveness of RMSNorm on heterogeneous archi-
tectures and initializations, covering different RNN variants and self-attentional
models, and various activations (such as sigmoid, tanh, linear and softmax), with
initializations ranging from uniform, normal, orthogonal with different initial-
ization ranges or variances. Details can be found in previous work on which we
base our comparisons, but we will include more detail to be self-contained.

Figure 1: BLEU curve over training steps on new-stest2013 devset.

In addition, we also experiment on the CIFAR-10 classification task. We train
a modified version of the ConvPool-CNN-C architecture, and follow the same
experimental protocol as in the WeightNorm paper [20] using their public source
code. LayerNorm is applied to the width and height dimensions of image rep-
resentation. We perform gain scaling and bias shifting on the channel dimension.
Our results (Table 1) show that RMSNorm outperforms Baseline and LayerNorm
in test error, and achieves 15% speed-up over LayerNorm, though it underper-
forms the BatchNorm and WeightNorm.

| Model | Test Error | Time |
|---|---|---|
| Baseline | 8.96% | 51s |
| BatchNorm | 8.25% | 66s |
| WeightNorm | 8.28% | 53s |
| LayerNorm | 10.49% | 72s |
| RMSNorm | 8.83% | 61s (**15%**) |

Table 1: Test error and time (sec) per train-ing epoch on CIFAR-10 classification task. The speedup of RMSNorm over LayerNorm is shown in bracket.

*Comparison with weight normalization:* We performed experiments with RNNSearch, using the WeightNorm im-
plementation provided by the base toolkit (Theano-version Nematus). Results in Figure 1 show that WeightNorm
converges slower and requires more training steps. In addition, the overall translation quality of WeightNorm on testsets
(21.7/23.5 on Test14/Test17, respectively) underperforms those of LayerNorm and (p)RMSNorm. We also attempted
integrating WeightNorm into pytorch-based RNNSearch using the official API (*nn.utils.weight_norm*), but this led to
out-of-memory problems.

= *To R3:* We will include the recent discussion on internal covariate shift in our final version. The scalar notation in (2)
follows LayerNorm paper [3], and we will change (1) to make the whole paper consistent. By "1%" in Fig 3, it actually
means 10%. In Table 7, "OE[30]" denotes the original results reported by [30]. [3] reproduce their work ("OE[3]"), and
add LayerNorm ("OE+LayerNorm[3]") to demonstrate LayerNorm's effectiveness. All these numbers are from existing
work, and other numbers are from our own experiments. We will make this clear in our final version.

= *To R4:* Please see the above common response.

= *To R5: On $l_p$ norm:* We didn't experiment with all choices of $p$ for $l_p$ norm, but we experimented with $l_2$ norm for
RNNSearch. Results in Fig. 2 and Table 2 show that L2Norm does not work well in terms of both convergence and
final translation quality.

*On optimizer hyperparameters:* For NMT model, we adopt Adam optimizer. The RNNSearch model is trained with
an initial learning rate of $10^{-4}$, which is half-decayed if no improvement is observed on devset. The learning rate for
Transformer is adapted according to Eq. (3) in paper [29] with a warmup step of 4000. We adopt the base setting. We
will include these details in the final version.

*On mean-centering and weight initialization:* See common response for the range
of weight initializations tested; R5 suggests that mean-centering in LayerNorm
(which RMSNorm abandons) may make models more robust towards arbitrary
weight/bias initializations. We perform an experiment on RNNSearch MT model
with tensorflow-Nematus, and change the center of weight initialization to 0.2.
Results in Figure 2 show that LayerNorm becomes very unstable with abnormal
initialization, but RMSNorm is more robust (both underperform the original
initialization). Our empirical evidence so far suggests that RMSNorm is similarly
robust as LayerNorm, or more.

Table 2: BLEU curve of LayerNorm and RM-SNorm on devset when initialization center is around 0.2.

*c. Error bars for reported accuracies and timing numbers* We perform only a single full training run for each of the
≈ 30 models due to resource limitations. Note that we *do not* claim RMSNorm is better than LayerNorm in quality,
but *comparable*. For the timing numbers, we report the standard deviation of three runs on three different models
(for Baseline/LayerNorm/RMSNorm, respectively): 3.4/32.5/11.8 (RNNSearch with tensorflow-Nematus), 6.3/5.7/5.2
(Attentive Reader model) and 0.23/1.31/0.035 (Transformer model; extremely low variance due to use of different
computing platform). We will show more details in the final version.

[Meta-Review · NeurIPS 2019]

The authors present a new form of normalization for deep networks called RMSNorm. This normalization acts like layer normalization but without mean centering. Because the method only requires a single pass of statistics calculations, the authors demonstrate improved training times for both machine translation and image caption retrieval while maintaining predictive accuracy. As commented by the reviewers, the paper is clearly written; the results are clearly presented and the experiments are quite thorough (different ML systems; ML architectures). In sum, the results and convincing (1 reviewer upgrade their score accordingly) and the results are use-able by those that build language models and potentially other forms of deep networks that require normalization schemes. For these reasons, assuming the authors revise the paper to address all reviewer comments, this paper is accepted to this conference.